# Factors Associated with Dietary Change since the Outbreak of COVID-19 in Japan

**DOI:** 10.3390/nu13062039

**Published:** 2021-06-14

**Authors:** Misa Shimpo, Rie Akamatsu, Yui Kojima, Tetsuji Yokoyama, Tsuyoshi Okuhara, Tsuyoshi Chiba

**Affiliations:** 1Department of Food and Health Sciences, Faculty of Health and Human Development, The University of Nagano, Nagano City, Nagano 380-8525, Japan; 2Natural Science Division, Faculty of Core Research, Ochanomizu University, Bunkyo-ku, Tokyo 112-8610, Japan; akamatsu.rie@ocha.ac.jp; 3Department of Health and Nutrition, Faculty of Human Life Studies, University of Niigata Prefecture, Nagano City, Niigata 950-8680, Japan; y_kojima@unii.ac.jp; 4Department of Health Promotion, National Institute of Public Health, Wako-shi, Saitama 351-0197, Japan; yokoyama.t.aa@niph.go.jp; 5Department of Health Communication, School of Public Health, The University of Tokyo, Bunkyo-ku, Tokyo 113-8655, Japan; okuhara-ctr@umin.ac.jp; 6Department of Food Function and Labeling, National Institute of Health and Nutrition, National Institutes of Biomedical Innovation, Health and Nutrition, Shinjuku-ku, Tokyo 162-8636, Japan; tyschiba@nibiohn.go.jp

**Keywords:** dietary habits, lifestyle, COVID-19

## Abstract

In Japan, dietary habits have greatly changed since the coronavirus disease (COVID-19) outbreak; we examined factors related to dietary changes. An online cross-sectional questionnaire survey was conducted in November 2020 among 6000 Japanese adults (aged 20–64 years) registered with a research company and gathered data on demographics, socioeconomic factors, medical history, COVID-19 status of the respondent’s family and neighbors, fear of COVID-19, and changes in lifestyle and dietary habits since the COVID-19 outbreak. To the question “Have you made healthier changes to your dietary habits compared with the dietary habits before the spread of COVID-19 (1 year ago, November 2019)?”, 1215 (20.3%), 491 (8.2%), and 4294 (71.6%) participants answered that their dietary habits were healthier, unhealthier, and unchanged, respectively. Healthier and unhealthier dietary habits were associated with greater fear of COVID-19, altered exercise and sleep times, and smoking. Unhealthy habits were positively associated with living alone, decreasing household income, colleagues with COVID-19, stress, and weight loss/gain. Annual household income, changing household income, COVID-19 in friends, health literacy, exercise frequency, weight loss, and starting smoking were positively associated with healthier dietary changes. The generalizability of these results and strategies to inculcate healthy diets in this “new normal” should be investigated.

## 1. Introduction

Our lifestyle, including diet, exercise habits, and work habits, needed to change to prevent the spread of the coronavirus disease (COVID-19). Many countries and territories worldwide have enforced community-wide lockdowns, home quarantines, work from home, and social distancing since the outbreak of COVID-19. In Japan, the government declared a state of emergency in seven prefectures on 7 April 2020, and a nationwide emergency was imposed on 16 April 2020 [1]. The nationwide state of emergency ended on 25 May 2020 [1]. Thereafter, the adoption of a “new normal,” which covers shopping, leisure and sports, meals, use of public transportation, and participation in events, was required to prevent the spread of COVID-19 [2]. The Subcommittee on Novel Coronavirus Disease Control in Japan proposed five situations that pose a high risk of infection, which include social gatherings involving alcohol consumption, long-duration meals in large groups, conversing without a mask, living together in a small, limited space, and changing one’s location [3]. It is likely that dietary habits have changed greatly to avoid the risk of infection during mealtimes.

Many studies have investigated the changes in dietary habits since the onset of the COVID-19 pandemic; however, the results have varied. Some studies reported that the percentage of participants who were eating more was 34.3% in Poland [4] and 36.3% in Spain [5]. However, other studies reported that the proportion of those who increased their food intake was 43.5% in Poland [6], 11.4% in Spain, 18.1% in Greece [7], and 51.3% in Chile [8]. Furthermore, the proportion of participants who reduced food intake was 74.3% in Spain, 63.1% in Greece [7], and 14.9% in Chile [8]. Some studies investigated whether eating habits were healthier or unhealthier than before the COVID-19 pandemic. Eating habits became healthier or improved in 9.6% of participants in the Netherlands [9], 11.6% of participants in Spain, 21.1% of participants in Greece [7], 23% of participants in China [10], and 33.7% of participants in Chile [8]. However, eating habits became unhealthier or worsened in 7.1% of subjects in the Netherlands [9], 36.1% of subjects in Spain, 33.7% of subjects in Greece [7], and 26.7% of subjects in Chile [8]. Previous studies suggested that the proportion of those who did not undertake healthy changes in dietary habits was approximately 40–80% (39.6% in Chile [8], 52.3% in Spain, 45.2% in Greece [7], and 83.3% in the Netherlands) [9]. With regard to specific changes in dietary habits, the consumption of vegetables [4,8,10,11,12], fruits [4,8,10,11,12], milk products [10,11,12], snacks [4,5,10,11,12], fast foods [5,11], and alcohol [4,5,11] increased or decreased. Moreover, some studies reported that the frequency of home cooking increased during lockdowns [5,8]. Evidently, dietary habits have become either healthier or unhealthier. Thus, it is necessary to investigate the factors related to the differences in dietary changes.

However, the results of previous studies that investigated factors related to dietary changes are inconsistent. For example, some studies reported that age was negatively associated with healthy dietary change [4,13], whereas another study reported that age was negatively associated with increased intake of junk food [11]. Federico et al. reported that older people did not change their dietary habits [9]. The findings on weight and educational level were similar to those on age, with reports of both positive and negative associations between these factors and dietary changes [4,6,9,11,13]. Moreover, some studies suggested that residential area and sex were related to dietary change [5,9]. These abovementioned studies were conducted in various countries.

Although studies on dietary changes have been conducted worldwide, only a few studies were conducted in Japan [14]. Japan was one of the countries that had a less stringent governmental response to COVID-19 [15,16]. The Japanese government did not declare a lockdown and, instead, issued stay-at-home requests and relied on voluntary compliance. The previous studies on dietary changes since the outbreak of COVID-19 were conducted in Europe, China, and America. As the governmental responses to COVID-19 varied by country, it is necessary to investigate the dietary changes and the associated factors in Japan. To formulate future strategies to promote healthy diets during and after the COVID-19 pandemic, it is important to elucidate the factors related to dietary changes. In addition, dietary guidelines are not homogeneous across regions, and each country utilizes specific icons to describe specific food recommendations, such as a healthy eating patterns, food compositions, and food portions, through depictions that include pyramids, plates, and other illustrations [17]. It is necessary to develop strategies that fit the needs of the populations in different countries. Therefore, this study aimed to examine the factors related to dietary changes since the spread of COVID-19. It was hypothesized that dietary changes were influenced by demographic, sociopsychological, and structural variables and perceived threat based on the health belief model [18] (Figure 1). In addition, we investigated lifestyle habits and changes, such as exercise, sleep, and smoking, which were expected to be related to dietary habits.

## 2. Materials and Methods

### 2.1. Participants and Procedures

An online cross-sectional questionnaire survey was conducted in November 2020. The participants were 6000 Japanese adults (aged 20–64 years) who were registered with a research company, MyVoice Communications, Inc. We referred to the results of previous studies to calculate the sample size and estimated that the ratio of those who changed to a healthier diet, those who changed to an unhealthier diet, and those who did not change was 1:1:3 [6,11]. The results of the sample size calculation for a power of 0.8 and statistical significance level of 5% suggested that the number of people who changed to a healthier or unhealthier diet was 1128 each and the number of people who did not change was 3382. Considering invalid respondents, the estimated sample size was defined as comprising 6000 participants.

The research company delivered an invitation for participation in the questionnaire survey to 35,970 registered potential participants. Figure 2 is a flowchart of the study’s process flow and clarifies the inclusion and exclusion of participants in the study. From November 6 to 12, 2020, 8941 registered potential participants answered the questionnaire. The questionnaire informed the potential participants that the survey was for research purposes and that they could refuse to answer any question. After providing voluntary consent to participate, the participants answered the questionnaire. Among the 8941 respondents, 7482 respondents followed the instructional manipulation check, which detects participants who are not following instructions, and 1459 respondents did not follow it and excluded, thereby improves the quality of survey responses. This method was validated by previous studies [19,20]. Seventy-five respondents were excluded from these 7482 respondents because the research company judged that they did not answer the questionnaire accurately. Subsequently, 6000 participants were chosen randomly from the 7407 respondents in accordance with the population composition of prefectures [21], considering deviations of age and sex. This study was approved by the Ethics Committee of the University of Nagano (authorization number: E20-3).

### 2.2. Measurements

The questionnaire included demographic, sociopsychological, and structural variables and fear of COVID-19, lifestyle habits and changes, and the changes in dietary habits from before the spread of COVID-19 (November 2019 to November 2020). Some items of the questionnaire were based on previous studies [22,23,24,25,26,27] whereas other items were developed by three researchers who are registered dietitians in Japan, and these experts in health communication, nutrition science, and statistics confirmed the validity of the questionnaire. Furthermore, a technical official in the Department of Nutrition at the Ministry of Health, Labor and Welfare validated the questionnaire.

The demographic variables included age, sex, marital status, residential area, living alone, living with children (<15 years old), and living with elderly people (>65 years old). The sociopsychological variables included educational background, annual household income (2019), household income change (October 2019 to October 2020), and the business sector (fishery, agriculture, and forestry (fishery, agriculture), construction, manufacture, electricity, gas, and heat supply and water (electricity, gas), information and communication (information), transportation, wholesale and retail trade (wholesale), finance, insurance, and real estate lessors (finance), accommodation and eating and drinking services (accommodation), medical services and social welfare (medical services), education and learning support (school education), cooperatives and postal services (compound services), other services except for cooperatives and postal services (other services), government and local public services (public services), other, and not working) [22]. The structural variables included medical history, COVID-19 infection, health literacy (using the 14-item health literacy scale for Japanese adults (HLS-14) [23], Cronbach’s alpha = 0.864), stress (nine items included with three sub-scales, namely fatigue, anxiety, and depression [24]; Cronbach’s alpha = 0.948), height, and weight from before the declaration of the first emergency for Japan (April 2020) and the values recorded in November 2020. The fear of COVID-19 was determined using the seven items of the fear of COVID-19 scale and was evaluated on a 5-point Likert scale (1 = “strongly disagree” to 5 = “strongly agree”) [25]; Cronbach’s alpha was 0.866 for the fear of COVID-19. A higher score indicated greater fear of COVID-19. The items of lifestyle habits included frequency of exercise, sleep time, and smoking. The item of smoking status included the smoking status in November 2019 and in November 2020. These abovementioned items were selected in reference to the survey by the Ministry of Health, Labor and Welfare [26,27]. The items of lifestyle change included changes of exercise times per week and sleep duration per day based on answers of “decrease,” “increase,” and “no change.” The items of change in dietary habits before the spread of COVID-19 included the following question: “Have you made healthier changes to your dietary habits compared with the dietary habits before the spread of COVID-19 (1 year ago, November 2019)?” The following response options were available: “My dietary habits have become healthier,” “My dietary habits have become unhealthier,” and “My dietary habits have not changed.”

### 2.3. Statistical Analysis

Age was categorized as follows: 20–29, 30–39, 40–49, 50–59, and 60–64 years. Residential area was categorized into eight areas: North—Hokkaido; North-east—Tohoku (included Aomori, Iwate, Miyagi, Akita, Yamagata, and Fukushima prefectures); Eastern—Kanto (included Ibaraki, Tochigi, Gunma, Saitama, Chiba, Tokyo, and Kanagawa prefectures); Middle—Chubu (included Yamanashi, Nagano, Niigata, Toyama, Ishikawa, Fukui, Shizuoka, Aichi, Gifu, and Mie prefectures); Midwest—Kinki (included Shiga, Kyoto, Osaka, Hyogo, Nara, and Wakayama prefectures); Western—Chugoku (included Tottori, Shimane, Okayama, Hiroshima, and Yamaguchi prefectures); Southwest—Shikoku (included Kagawa, Ehime, Tokushima, and Kochi prefectures); and Southern—Kyushu (included Fukuoka, Saga, Nagasaki, Kumamoto, Oita, Miyazaki, Kagoshima, and Okinawa prefectures). Among the 6000 participants, 19 provided the same values for height and weight and 842 indicated that they did not know their annual household incomes. Weight was considered as the missing value if the values for weight and height were the same. Body mass index (BMI) was calculated based on weight and height. A weight change was defined as an “increase” or “decrease” based on a >5% difference in the weight from before the declaration of the first emergency in Japan (April 2020) to the weight recorded in November 2020. The annual household income was considered missing if it was unknown. The participants were divided into three groups: healthier, unhealthier, and unchanged groups, comprising participants whose responses were “My dietary habits have become healthier,” “My dietary habits have become unhealthier,” and “My dietary habits have not changed,” respectively. Categorical data are presented as number (percentage) and continuous data are presented as median (25th, 75th percentile). To analyze differences in health literacy, stress, fear of COVID-19, and BMI among the three groups, the Kruskal–Wallis analysis and Bonferroni’s multiple comparison test were conducted because the data were non-normally distributed. To analyze the differences in the distribution of other items among the three groups, the chi-square test was used. Furthermore, multinomial logistic regression analyses using stepwise methods were performed to analyze factors that influenced changes in dietary habits and to calculate the odds ratio (95% confidence interval (CI)) of healthier and unhealthier groups to the unchanged group. All analyses were conducted using IBM SPSS Statistics for Windows, version 26.0 (IBM Japan, Ltd., Tokyo, Japan). Statistical significance was set at *p* < 0.05.

## 3. Results

### 3.1. Demographic Characteristics

The 6000 respondents comprised 3044 (50.7%) males and 2956 (49.3%) females. The median (25, 75 percentile) age and BMI were 45 (34, 53) years and 21.5 (19.5, 24.1) kg/m^2^, respectively. The healthier, unhealthier, and unchanged groups comprised 1215 (20.3%), 491 (8.2%), and 4294 (71.6%) respondents, respectively.

### 3.2. Comparison of Demographic Variables among the Three Groups

Table 1 shows a comparison of demographic variables among the healthier, unhealthier, and unchanged groups. The distributions of age, marital status, living alone, living with children, and living with elderly people differed significantly among the three groups.

### 3.3. Comparison of Sociopsychological Variables among the Three Groups

Table 2 shows a comparison of sociopsychological variables among the healthier, unhealthier, and unchanged groups. The distributions of educational background, annual household income, and household-income change differed significantly among the three groups.

### 3.4. Comparison of Structural Variables among the Three Groups

Table 3 shows a comparison of structural variables among the healthier, unhealthier, and unchanged groups. The distributions of medical history of high blood pressure, dyslipidemia, stroke, chronic renal failure, and COPD, categories of BMI, and weight change differed significantly among the three groups. The health literacy score of the healthier group (52 (45,58)) was higher than that of the unhealthier (49 (43,55)) and unchanged groups (50 (42,56)). The stress score of the healthier group (16 (11,21)) was higher than that of the unchanged group (15 (10,20)) and lower than that of the unhealthier group (20 (16,26)), whereas the stress score of the unhealthier group was higher than that of the unchanged group. The distributions of COVID-19 infections of oneself, family living together, other family and relatives, colleagues, and friends differed significantly among the three groups. The fear-of-COVID-19 scores of the healthier group (19 (15,22)) were higher than those of the unchanged group (18 (14,21)), and was higher in the unhealthier group (19 (15,23)) compared with that in the unchanged group.

### 3.5. Comparison of Lifestyle Habits and Changes among the Three Groups

Table 4 shows a comparison of lifestyle habits and changes among the three groups. The distributions of frequency of exercise, sleep time, smoking status, changes of exercise frequency per week, sleep time per day, and smoking status were significantly different between the three groups.

### 3.6. Multinomial Logistic Regression Analyses of the Factors Associated with Changes in the Dietary Habits

Table 5 shows the results of multinomial logistic regression analyses of the factors associated with changes in the dietary habits. Participants with higher annual household income, changing household income, COVID-19 infection of friends, higher health literacy, decreasing weight, higher fear of COVID-19, higher frequency of exercise, those who smoked, changing exercise and sleep time, and started smoking were more likely to be healthier compared with the unchanged group. Participants with characteristics of living alone, decreasing household income, COVID-19 infection of colleagues, higher stress, changing weight, higher fear of COVID-19, smoking, and changing exercise frequency and sleep duration were more likely to have unhealthier dietary habits compared with the unchanged group.

## 4. Discussion

This study examined the factors related to dietary changes (healthier, unhealthier, and unchanged diets) since the spread of COVID-19. According to adjusted analysis, there were common factors associated with dietary changes, such as higher fear of COVID-19, changes in exercise frequency and sleep duration, and smoking. Participants who changed to healthier diets had higher annual household incomes, changing household income, COVID-19 infection of friends, higher health literacy, decreasing weight, and higher frequency of exercise, and had started smoking. Participants who changed to unhealthier diets were living alone and had decreasing household income, colleagues who had contracted COVID-19, higher stress, and weight change.

These factors, such as changing exercise frequency and sleep duration showed that those who changed dietary habits intended to change other lifestyle habits. It was assumed that those participants had sustained a greater impact of the COVID-19 pandemic compared with those who did not change their dietary habits. Healthcare providers are required to pay attention to individuals with a marked a change in lifestyle habits, because these habits have a great impact on the COVID-19 pandemic and could lead to changes in other lifestyle habits.

Of the common factors, fear of COVID-19 was a characteristic of factors associated with the dietary change caused by the COVID-19 pandemic. A previous study suggested that students with a greater fear of COVID-19 were more likely to continue smoking or drinking alcohol at unchanged or higher levels during the pandemic [28]. Another study suggested that fear of COVID-19 significantly mediated the associations between perceived health status and insomnia, mental health, and COVID-19-preventive behaviors [29]. The results of associations between dietary changes and fear of COVID-19 supported the previous studies and suggested that fear of COVID-19 was an important factor of individual changes from before COVID-19 pandemic.

Participants who made healthier dietary changes had a higher annual household income, health literacy, frequency of exercise, and weight loss. A previous study conducted in the USA suggested that a higher income was associated with larger changes in self-protective behaviors since the COVID-19 pandemic [30]. Moreover, people with high incomes may change dietary habits due to self-protective behavior. A previous study among Chinese adults reported that family annual income was positively associated with household dietary diversity score in COVID-19 lockdown [31]. According to a National Health and Nutrition Survey conducted in Japan in 2018, people with an annual household income of more than six million JPY had a higher frequency of eating well-balanced diets compared with those with an annual household income under two million JPY [26]. Therefore, participants who made healthier changes to their dietary habits and had higher annual household incomes may have practiced healthy diets before the spread of COVID-19. It is possible that the gap between those who practice healthy and unhealthy dietary habits is widening since the spread of COVID-19.

Contrary to expectations, more people started smoking among those who made healthier changes to their dietary habits. A previous study reported that 28% of tobacco users reported increasing their cigarette use during the pandemic; the reasons were boredom, stress/anxiety, and working from home [32]. Other studies reported that the reasons for greater consumption of nicotine were boredom, lack of social contacts, reward after a hard-working day, and loss of daily structure [33]. It can be assumed that those who made healthier changes to their dietary habits selected smoking as one of methods for stress reduction, in contrast with those who changed their dietary habits unhealthily and selected eating unhealthy food as one of methods for stress reduction. In addition, smoking was one of the common factors of dietary changes for healthier and unhealthier habits. One of the reasons for those who changed dietary habits unhealthily to smoking may be that they had higher stress levels. Smoking is one of the risks for developing severe COVID-19 [34]. It can be assumed that some smokers were afraid of becoming severely ill with COVID-19 and improved their diets. These considerations have not been proven as yet. There were a few people who changed their smoking habits in this study. Future studies need to investigate the relationship between change in dietary habits and smoking.

Participants who changed their dietary habits unhealthily were living alone and had higher stress levels. A systematic review reported that the COVID-19 pandemic and the lockdown measures caused psychological distress [35]. It is difficult for people who live alone to receive social support because the “new normal” recommends that we refrain from going and eating out and encourages working from home [2]. It is suggested that support is needed for people who are living alone for them to not change dietary habits unhealthily.

These results of the factors related to changes in dietary habits differed from those of previous studies, wherein age was also negatively associated with healthy and unhealthy dietary changes [4,11,13]. Moreover, older people tended not to change their dietary habits [9]. Some studies reported that the changes in dietary habits among obese people were healthy [4,13] and yet others found the dietary changes to be unhealthy [6,9,11]. In this study, age and weight were related to dietary change in the simple analysis, but there were no associations in multivariable analysis. Future studies are needed to clarify the factors associated with dietary changes in various countries.

This study has several limitations. First, this study could not be representative of the entire Japanese adult population because this survey was conducted online and involved people who registered with a single research company. However, the distribution of participants by prefecture in this study was close to the population distribution in Japan. This is because respondents were selected in accordance with the population composition of prefectures, considering deviation in age and sex. Second, we could not investigate all factors associated with dietary changes. Third, responses were only obtained at one time point because this study was conducted as a cross-sectional survey. Responses about changes were self-reported and based on the participant’s memory. Fourth, this study focused on general (nonspecific) dietary changes. Specific dietary changes should be investigated in future studies. In addition, it is necessary to investigate the long-term effects of the spread of COVID-19 on dietary habits.

## 5. Conclusions

This study identified factors related to dietary changes since the spread of COVID-19 in Japanese adults. Dietary habits became healthier, unhealthier, or remained unchanged. Some factors, such as higher fear of COVID-19, changing exercise and sleep times, and smoking were common factors between participants with healthy and unhealthy dietary changes. Those who changed to healthier diets had higher annual household incomes, changing household income, COVID-19 infection of friends, higher health literacy, decreasing weight, higher frequency of exercise, and started smoking. Those who changed to unhealthier diets were living alone, and had decreasing household income, COVID-19 infection of colleagues, higher stress, and weight change. For these results to be generalizable, further research is needed. The changes in lifestyle caused by the COVID-19 pandemic will persist for a while. Therefore, it is necessary to consider strategies to practice healthy diets in this “new normal.”

## Figures and Tables

**Figure 1 nutrients-13-02039-f001:**
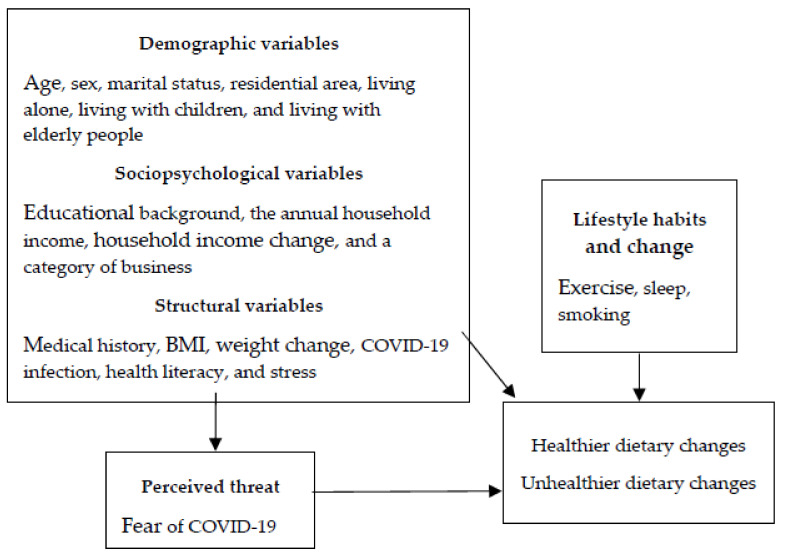
Hypothesis of this study.

**Figure 2 nutrients-13-02039-f002:**
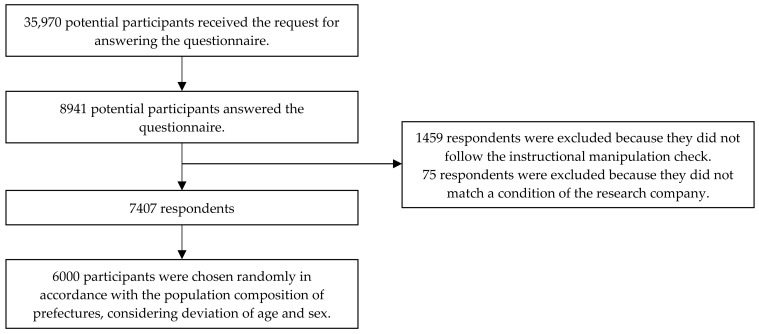
A flowchart of this study.

**Table 1 nutrients-13-02039-t001:** Comparison of demographic variables among the healthier, unhealthier, and unchanged groups.

Demographic Variables	Total *N* = 6000	Participants with Changes in Dietary Habits, *n* (%)			*p* ^1^
		Healthier Group *n* = 1215	Unhealthier Group *n* = 491	Unchanged Group *n* = 4294	
Sex					
Male	3044 (50.7)	584 (48.1)	242 (49.3)	2218 (51.7)	0.070
Female	2956 (49.3)	631 (51.9)	249 (50.7)	2076 (48.3)	
Age, years					
20–29	1099 (18.3)	250 (20.6)	120 (24.4)	729 (17.0)	<0.001
30–39	1256 (20.9)	246 (20.2)	108 (22.0)	902 (21.0)	
40–49	1602 (26.7)	300 (24.7)	127 (25.9)	1175 (27.4)	
50–59	1405 (23.4)	298 (24.5)	103 (21.0)	1004 (23.4)	
60–64	638 (10.6)	121 (10.0)	33 (6.7)	484 (11.3)	
Marital status					
Unmarried	2593 (43.2)	450 (37.0)	265 (54.0)	1878 (43.7)	<0.001
Married	3090 (51.5)	708 (58.3)	193 (39.3)	2189 (51.0)	
Divorced or widowed ^2^	317 (5.3)	57 (4.7)	33 (6.7)	227 (5.3)	
Residential area ^3^					
North region	240 (4.0)	47 (3.9)	15 (3.1)	178 (4.1)	0.075
North-east region	391 (6.5)	86 (7.1)	28 (5.7)	277 (6.5)	
Eastern region	2170 (36.2)	479 (39.4)	194 (39.5)	1497 (34.9)	
Middle region	1078 (18.0)	214 (17.6)	70 (14.3)	794 (18.5)	
Midwest region	971 (16.2)	175 (14.4)	86 (17.5)	710 (16.5)	
Western region	330 (5.5)	55 (4.5)	26 (5.3)	249 (5.8)	
Southwest region	167 (2.8)	28 (2.3)	13 (2.6)	126 (2.9)	
Southern region	653 (10.9)	131 (10.8)	59 (12.0)	463 (10.8)	
Living alone					
Yes	1076 (17.9)	186 (15.3)	139 (28.3	751 (17.5)	<0.001
No	4924 (82.1)	1029 (84.7)	352 (71.7)	3543 (82.5)	
Living with children (<15 years old)					
Yes	1173 (19.6)	279 (23.0)	70 (14.3)	824 (19.2)	<0.001
No	4827(80.5)	936 (77.0)	421 (85.7)	3470 (80.8)	
Living with elderly people (>65 years old)					
Yes	1557 (26.0)	296 (24.4)	109 (22.2)	1152 (26.8)	0.032
No	4443 (74.1)	919 (75.6)	382 (77.8)	3142 (73.2)	

^1^ χ^2^ test; ^2^ Divorced or widowed; ^3^ North—Hokkaido, North-east—Tohoku, Eastern—Kanto, Middle—Chubu, Midwest—Kinki, Western—Chugoku, Southwest—Shikoku, Southern—Kyushu in Japanese.

**Table 2 nutrients-13-02039-t002:** Comparison of sociopsychological variables among the healthier, unhealthier, and unchanged groups.

Sociopsychological Variables	Total *N* = 6000	Participants with Changes in Dietary Habits, *n* (%)			*p* ^1^
		Healthier Group *n* = 1215	Unhealthier Group *n* = 491	Unchanged Group *n* = 4294	
Educational background					
Junior high school	132 (2.2)	16 (1.3)	11 (2.2)	105 (2.4)	<0.001
High school	1518 (25.3)	253 (20.8)	123 (25.1)	1142 (26.6)	
Professional school	682 (11.4)	116 (9.5)	59 (12.0)	507 (11.8)	
Junior college ^2^	616 (10.3)	136 (11.2)	44 (9.0)	436 (10.2)	
College ^3^	2702 (45.0)	606 (49.9)	230 (46.8)	1866 (43.5)	
Graduate school	350 (5.8)	88 (7.2)	24 (4.9)	238 (5.5)	
Industrial classification					
Fishery, agriculture	44 (0.7)	9 (0.7)	4 (0.8)	31 (0.7)	0.184
Construction	204 (3.4)	38 (3.1)	22 (4.5)	144 (3.4)	
Manufacture	743 (12.4)	154 (12.7)	71 (14.5)	518 (12.1)	
Electricity, gas	53 (0.9)	12 (1.0)	5 (1.0)	36 (0.8)	
Information	274 (4.6)	66 (5.4)	25 (5.1)	183 (4.3)	
Transportation	207 (3.5)	28 (2.3)^-^	24 (4.9)	155 (3.6)	
Wholesale	475 (7.9)	96 (7.9)	32 (6.5)	347 (8.1)	
Finance	237 (4.0)	52 (4.3)	16 (3.3)	169 (3.9)	
Accommodations	130 (2.2)	20 (1.6)	10 (2.0)	100 (2.3)	
Medical services	397 (6.6)	73 (6.0)	33 (6.7)	291 (6.8)	
School education	265 (4.4)	76 (6.3)	22 (4.5)	167 (3.9)^-^	
Compound services	36 (0.6)	9 (0.7)	3 (0.6)	24 (0.6)	
Other services	564 (9.4)	106 (8.7)	47 (9.6)	411 (9.6)	
Public services	256 (4.3)	55 (4.5)	17 (3.5)	184 (4.3)	
Other	530 (8.8)	110 (9.1)	41 (8.4)	379 (8.8)	
Not working	1585 (26.4)	311 (25.6)	119 (24.2)	1155 (26.9)	
Annual household income (million JPY) ^4^					
Less than one	309 (6.0)	48 (4.4)	38 (8.8)	223 (6.1)	<0.001
One to two	353 (6.8)	45 (4.2)	39 (9.0)	269 (7.4)	
Two to three	540 (10.5)	93 (8.6)	65 (15.1)	382 (10.5)	
Three to four	663 (12.9)	136 (12.5)	71 (16.5)	456 (12.5)	
Four to five	655 (12.7)	116 (10.7)	45 (10.4)	494 (13.6)	
Five to six	565 (11.0)	112 (10.3)	40 (9.3)	413 (11.3)	
Six to seven	489 (9.5)	113 (10.4)	41 (9.5)	335 (9.2)	
Seven to eight	431 (8.4)	87 (8.0)	27 (6.3)	317 (8.7)	
Eight to nine	277 (5.4)	76 (7.0)	15 (3.5)	186 (5.1)	
Nine to ten	261 (5.1)	75 (6.9)	11 (2.6)	175 (4.8)	
More than ten	615 (11.9)	183 (16.9)	39 (9.0)	393 (10.8)	
Household-income change					
Decrease	1317 (22.0)	343 (28.2)	175 (35.6)	799 (18.6)	<0.001
Increase	362 (6.0)	104 (8.6)	32 (6.5)	226 (5.3)	
No change	4321 (72.0)	768 (63.2)	284 (57.8)	3269 (76.1)	

^1^ χ^2^ test; ^2^ Junior colleges or technical colleges; ^3^ College or university; ^4^ one million JPY = 9515 USD (15 February 2021, according to Yahoo! Japan finance); missing data of the annual household income, *n* = 842 respondents.

**Table 3 nutrients-13-02039-t003:** Comparison of structural variables among the healthier, unhealthier, and unchanged groups.

Structural Variables	Total *N* = 6000	Participants with Changes in Dietary Habits, *n* (%)			*p* ^1^
		Healthier Group *n* = 1215	Unhealthier Group *n* = 491	Unchanged Group *n* = 4294	
Medical history					
Diabetes					
No	5789 (96.5)	1172 (96.5)	465 (94.7)	4152 (96.7)	0.077
Yes	211 (3.5)	43 (3.5)	26 (5.3)	142 (3.3)	
High blood pressure					
No	5408 (90.1)	1077 (88.6)	434 (88.4)	3897(90.8)	0.037
Yes	592 (9.9)	138 (11.4)	57 (11.6)	397 (9.2)	
Dyslipidemia					
No	5524 (92.1)	1099 (90.5)	432 (88.0)	3993 (93.0)	<0.001
Yes	476 (7.9)	116 (9.5)	59 (12.0)	301 (7.0)	
Heart disease					
No	5899 (98.3)	1190 (97.9)	480 (97.8)	4229 (98.5)	0.260
Yes	101 (1.7)	25 (2.1)	11 (2.2)	65 (1.5)	
Stroke					
No	5943 (99.1)	1195 (98.4)	483 (98.4)	4265 (99.3)	0.002
Yes	57 (1.0)	20 (1.6)	8 (1.6)	29 (0.7)	
Chronic renal failure					
No	5962 (99.4)	1205 (99.2)	481 (98.0)	4276 (99.6)	<0.001
Yes	38 (0.6)	10 (0.8)	10 (2.0)	18 (0.4)	
COPD ^1^					
No	5974 (99.6)	1201 (98.8)	488 (99.4)	4285 (99.8)	<0.001
Yes	26 (0.4)	14 (1.2)	3 (0.6)	9 (0.2)	
Malignant neoplasms					
No	5842 (97.4)	1172 (96.5)	476 (96.9)	4194 (97.7)	0.056
Yes	158 (2.6)	43 (3.5)	15 (3.1)	100 (2.3)	
BMI (kg/m^2^)					
<18.5	841 (14.1)	149 (12.3)	78 (15.9)	614 (14.4)	0.001
18.5–25.0	3986 (66.6)	849 (70.0)	292 (59.5)	2845 (66.5)	
>25.0	1154 (19.3)	215 (17.7)	121 (24.6)	818 (19.1)	
Weight change ^2^					
Decrease	308 (5.1)	118 (9.7)	44 (9.0)	146 (3.4)	<0.001
Increase	321 (5.4)	79 (6.5)	77 (15.7)	165 (3.9)	
No change	5352 (89.5)	1016 (83.8)	370 (75.4)	3966 (92.7)	
Median value (25, 75 percentile)					*p* ^3^
Health literacy	50 (43, 56)	52 (45, 58) ^a^	49 (43, 55) ^b^	50 (42, 56) ^b^	<0.001
Stress	16 (11, 21)	16 (11, 21) ^a^	20 (16, 26) ^b^	15 (10, 20) ^c^	<0.001
COVID-19 infection					
Oneself	*n* (%)				*p* ^1^
Not infected	5958 (99.3)	1189 (97.9)	484 (98.6)	4285 (99.8)	<0.001
Infected	42 (0.7)	26 (2.1)	7 (1.4)	9 (0.2)	
Family living together					
Not infected	5968 (99.5)	1197 (98.5)	487 (99.2)	4284 (99.8)	<0.001
Infected	32 (0.5)	18 (1.5)	4 (0.8)	10 (0.2)	
Other family and relatives					
Not infected	5957 (99.3)	1193 (98.2)	485 (98.8)	4279 (99.7)	<0.001
Infected	43 (0.7)	22 (1.8)	6 (1.2)	15 (0.3)	
Colleagues					
Not infected	5822 (97.0)	1159 (95.4)	462 (94.1)	4201 (97.8)	<0.001
Infected	178 (3.0)	56 (4.6)	29 (5.9)	93 (2.2)	
Friends					
Not infected	5863 (97.7)	1152 (94.8)	475 (96.7)	4236 (98.6)	<0.001
Infected	137 (2.3)	63 (5.2)	16 (3.3)	58 (1.4)	
Median value (25, 75 percentile)					*p* ^3^
Fear of COVID-19	18 (14, 21)	19 (15, 22) ^a^	19 (15, 23) ^a^	18 (14, 21) ^b^	<0.001

^1^ χ^2^ test; ^2^ “increase” included a >5% weight gain and “decrease” included a >5% weight loss from the weight recorded in April 2020, ^3^ Kruskal-Wallis test; ^a,b,c^ Bonferroni’s multiple comparison test results with significant differences (adjusted *P* < 0.05); A higher score indicated greater fear of COVID-19.

**Table 4 nutrients-13-02039-t004:** Comparison of lifestyle habits among the healthier, unhealthier, and unchanged groups.

Lifestyle Habits	Total *N* = 6000	Participants with Changes in Dietary Habits, *n* (%)			*p* ^1^
		Healthier Group *n* = 1215	Unhealthier Group *n* = 491	Unchanged Group *n* = 4294	
Exercise frequency					
Almost none	3386 (56.4)	436 (35.9)	305 (62.1)	2645 (61.6)	<0.001
1–3 times a month	437 (7.3)	95 (7.8)	46 (9.4)	296 (6.9)	
1–2 times a week	937 (15.6)	270 (22.2)	64 (13.0)	603 (14.0)	
3–4 times a week	545 (9.1)	177 (14.6)	39 (7.9)	329 (7.7)	
5–6 times a week	353 (5.9)	112 (9.2)	18 (3.7)	223 (5.2)	
Every day	342 (5.7)	125 (10.3)	19 (3.9)	198 (4.6)	
Sleep time, hours					
<5	580 (9.7)	88 (7.2)	83 (16.9)	409 (9.5)	<0.001
5–6	1789 (29.8)	349 (28.7)	149 (30.3)	1291 (30.1)	
6–7	2056 (34.3)	449 (37.0)	127 (25.9)	1480 (34.5)	
7–8	1174 (19.6)	261 (21.5)	90 (18.3)	823 (19.2)	
8–9	282 (4.7)	52 (4.3)	23 (4.7)	207 (4.8)	
>9	119 (2.0)	16 (1.3)	19 (3.9)	84 (2.0)	
Smoking					
No	4903 (81.7)	956 (78.7)	392 (79.8)	3555 (82.8)	0.003
Yes	1097 (18.3)	259 (21.3)	99 (20.2)	739 (17.2)	
Lifestyle changes					
Exercise times per week					
Decreased	1094 (18.2)	320 (26.3)	223 (45.4)	551 (12.8)	<0.001
Increased	642 (10.7)	310 (25.5)	50 (10.2)	282 (6.6)	
No change	4264 (71.1)	585 (48.1)	218 (44.4)	3461 (80.6)	
Sleep time per day					
Decreased	588 (9.8)	146 (12.0)	144 (29.3)	298 (6.9)	<0.001
Increased	717 (12.0)	341 (28.1)	105 (21.4)	271 (6.3)	
No change	4695 (78.3)	728 (59.9)	242 (49.3)	3725 (86.7)	
Smoking					
Quit smoking	70 (1.2)	24 (2.0)	10 (2.0)	36 (0.8)	<0.001
Started smoking	25 (0.4)	16 (1.3)	3 (0.6)	6 (0.1)	
No change ^2^	5905 (98.4)	1175 (96.7)	478 (97.4)	4252 (99.0)	

^1^ χ^2^ test, ^2^ No change included providing the same response, either “smoking” or “not smoking,” at both time points, in November 2019 and November 2020, in this study.

**Table 5 nutrients-13-02039-t005:** Multinomial logistic regression analyses of the factors associated with change in dietary habits.

Factors	Reference	Participants with Changes in Dietary Habits Odds Ratio (95% CI) Based on Unchanged Group	
		Healthier Group	Unhealthier Group
Demographic variables			
Living alone	Not living alone	0.93 (0.76–1.15)	1.62 (1.25–2.10) ***
Sociopsychological variables			
Annual household income		1.08 (1.05–1.10) ***	0.97 (0.93–1.01)
Household-income change			
Decrease	No change	1.42 (1.18–1.70) ***	1.53 (1.20–1.95) **
Increase	No change	1.58 (1.18–2.10) **	1.09 (0.70–1.69)
Structural variables			
Colleague	Uninfected	1.39 (0.92–2.10)	1.88 (1.13–3.14) *
Friends	Uninfected	2.05 (1.32–3.18) **	1.54 (0.81–2.91)
Health literacy		1.02 (1.01–1.03) ***	1.00 (0.98–1.01)
Stress		0.99 (0.98–1.00)	1.05 (1.04–1.07) ***
Weight change			
Decrease	No change	2.32 (1.72–3.13) ***	2.21 (1.47–3.32) ***
Increase	No change	1.28 (0.92–1.78)	3.11 (2.21–4.37) ***
Perceived threat			
Fear of COVID-19		1.06 (1.04–1.07) ***	1.02 (1.00–1.04) *
Lifestyle habits			
Exercise		1.22 (1.16–1.28) ***	0.96 (0.88–1.04)
Smoking	Not smoking	1.43 (1.18–1.73) ***	1.37 (1.04–1.80) *
Lifestyle change			
Exercise frequency per week			
Decrease	No change	2.22 (1.81–2.72) ***	2.88 (2.19–3.78) ***
Increase	No change	3.10 (2.45–3.92) ***	1.94 (1.29–2.92) **
Sleep duration per day			
Decrease	No change	1.44 (1.11–1.87) **	3.10 (2.29–4.20) ***
Increase	No change	3.07 (2.46–3.83) ***	2.69 (1.95–3.71) ***
Smoking			
Quit smoking	No change	1.67 (0.88–3.17)	1.90 (0.84–4.29)
Started smoking	No change	4.25 (1.23–14.7) *	1.65 (0.31–8.77)

* *p* < 0.05, ** *p* < 0.01, *** *p* < 0.001, Independent variables: demographic variables (age, sex, marital status, residential area, living alone, living with children, and living with elderly people); sociopsychological variables (educational background, annual household income, household income change, and business category); structural variables (medical history, COVID-19 infection, health literacy, stress, BMI category, weight change); perceived threat (fear of COVID-19 score); and lifestyle habits and change (exercise, smoking, sleep time). The independent variables were selected using a stepwise method.

## Data Availability

The raw data are not publicly available due to restrictions ethical.

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
