# Peer review of "Factors Associated with Dietary Change since the Outbreak of COVID-19 in Japan"

_nutrients, 2021, doi:10.3390/nu13062039_

Round 1

Reviewer 1 Report

Thank you for the opportunity to review the study conducted by Shimpo et al which examined factors related to dietary changes since the COVID-19 outbreak in Japan. I have some suggestions for the authors before this manuscript can be considered for publication.

English revision by a native English speaker is needed.

Some directions for further investigations should be stated at the end of the abstract as well as in the conclusions section.

“change” is not an adequate keyword.

In my opinion, it would be important for the authors to refer in the Introduction to the main guidelines regarding healthy dietary habits:

Fernandez, M. L., Raheem, D., Ramos, F., Carrascosa, C., Saraiva, A., & Raposo, A. (2021). Highlights of current dietary guidelines in five continents. International Journal of Environmental Research and Public Health18(6), 2814.

Willett, W. (2021). Mediterranean Dietary Pyramid. International Journal of Environmental Research and Public Health18(9), 4568.

Serra-Majem, L., Tomaino, L., Dernini, S., Berry, E. M., Lairon, D., Ngo de la Cruz, J., ... & Trichopoulou, A. (2020). Updating the mediterranean diet pyramid towards sustainability: Focus on environmental concerns. International Journal of Environmental Research and Public Health17(23), 8758.

How was the questionnaire applied in this study constructed? Was it properly validated? The authors should explain this better in section 2.

In section 2, it would also be interesting to provide a flowchart with all the steps taken to carry out the present work.

In section 4, the authors should provide some more studies around the world to discuss their results.

Author Response

Responses to Reviewer 1’s Suggestions

Manuscript ID: nutrients-1243411

Title: Factors associated with dietary change since the outbreak of COVID-19 in Japan

Authors: Misa Shimpo *, Rie Akamatsu, Yui Kojima, Tetsuji Yokoyama, Tsuyoshi Okuhara, Tsuyoshi Chiba

Reviewer 1

Reviewer’s Comments to Authors

Answer

Thank you for the opportunity to review the study conducted by Shimpo et al which examined factors related to dietary changes since the COVID-19 outbreak in Japan. I have some suggestions for the authors before this manuscript can be considered for publication.

Thank you for your constructive comments. We have carefully revised the manuscript according to the comments and feedback that we received from all the reviewers and have marked the revisions in the main manuscript using the “Track Changes” function. However, the proofreading-related changes made by the language editor are not marked. Please see our responses to each of your comments below.

English revision by a native English speaker is needed.

The original manuscript was edited by Editage, which provides language-editing services. We have conveyed your feedback to the agency, and they have edited the revised manuscript accordingly. The certificate of English editing is included with the files in this submission.

Some directions for further investigations should be stated at the end of the abstract as well as in the conclusions section.

We have added details of further investigations in the abstract and the conclusions (L34-37, 430-436).

“change” is not an adequate keyword.

We deleted “change” from the keywords.

In my opinion, it would be important for the authors to refer in the Introduction to the main guidelines regarding healthy dietary habits:

Fernandez, M. L., Raheem, D., Ramos, F., Carrascosa, C., Saraiva, A., & Raposo, A. (2021). Highlights of current dietary guidelines in five continents. International Journal of Environmental Research and Public Health, 18(6), 2814.

Willett, W. (2021). Mediterranean Dietary Pyramid. International Journal of Environmental Research and Public Health, 18(9), 4568.

Serra-Majem, L., Tomaino, L., Dernini, S., Berry, E. M., Lairon, D., Ngo de la Cruz, J., ... & Trichopoulou, A. (2020). Updating the mediterranean diet pyramid towards sustainability: Focus on environmental concerns. International Journal of Environmental Research and Public Health, 17(23), 8758.

We added another reference citation that reports findings from a study on dietary guidelines in the last paragraph of the introduction (L96-100).

How was the questionnaire applied in this study constructed? Was it properly validated? The authors should explain this better in section 2.

We added an explanation about the questionnaire in the section 2.2 Measurements (L160-165).

In section 2, it would also be interesting to provide a flowchart with all the steps taken to carry out the present work.

We have included a flowchart of the study’s process flow in section 2.

In section 4, the authors should provide some more studies around the world to discuss their results.

Accordingly, we made changes to the analysis in response to this comment and the comment by reviewer 2 and have changed the results and discussion sections. We have discussed our new results and added details of some considerations and references as well (L345-350, 363-365, 374-378). Please check if these changes adequately address the concern.

In accordance with the official style, we revised the funding statement as follows: “This research was supported by the MHLW Special Research Program (Grant Number JPMH20CA2040).”

Reviewer 2 Report

Dear Authors,

Congratulation or your article, we let you some small suggestions for the final version.

Line 45, please put one citation for the affirmation “In particular, dietary habits have changed 45 greatly because the risk of infection is high when we remove our masks to eat.”

We are on the opinion that it was important to know if the physical exercise was not changed, or if it increased, or if it diminished. Without this information it is more difficult to interpret the effect of the pandemic situation on BMI. It can be a limit of your study?

Curiosity the smokers changed to healthier eating, have you some explanation for this?

Author Response

Responses to Reviewer 2’s Suggestions

Manuscript ID: nutrients-1243411

Title: Factors associated with dietary change since the outbreak of COVID-19 in Japan

Authors: Misa Shimpo *, Rie Akamatsu, Yui Kojima, Tetsuji Yokoyama, Tsuyoshi Okuhara, Tsuyoshi Chiba

Reviewer 2

Reviewer’s Comments to Authors

Answer

Dear Authors,

Congratulation or your article, we let you some small suggestions for the final version.

Thank you for your constructive comments. We have carefully revised the manuscript according to the comments and feedback that we received from all the reviewers and have marked the revisions in the main manuscript using the “Track Changes” function. However, the proofreading-related changes made by the language editor are not marked. Please see our responses to each of your comments below.

Line 45, please put one citation for the affirmation “In particular, dietary habits have changed greatly because the risk of infection is high when we remove our masks to eat.”

Thank you for this suggestion. We added a relevant reference citation and revised sentences (L49-54).

We are on the opinion that it was important to know if the physical exercise was not changed, or if it increased, or if it diminished. Without this information it is more difficult to interpret the effect of the pandemic situation on BMI. It can be a limit of your study?

Thank you for this comment. We examined the weekly changes in the exercise duration. In response to your feedback, we reanalyzed the study’s data and added information on the weekly changes in exercise duration, daily sleep duration, smoking, and changes in household income and weight in our analysis as these items were included in our study questionnaire. Due to the reanalysis, the results, discussion, conclusion, and abstract section have been modified. In addition, we added details of the limitation with regard to the items of change. Please check if these changes adequately address the concern.

Curiosity the smokers changed to healthier eating, have you some explanation for this?

This result has been changed in the revised manuscript because of the changes in the results of the reanalysis. However, there were associations between dietary changes and smoking. Accordingly, we have added the details of these considerations in the revised manuscript (L373-389).

In accordance with the official style, we revised the funding statement as follows: “This research was supported by the MHLW Special Research Program (Grant Number JPMH20CA2040).” This is the official style.

Round 2

Reviewer 1 Report

Thank you for addressing my suggestions.